# Health Knowledge About Early Diagnosis of Autism Spectrum Disorders: A Case for Soft Transdiagnostic Approaches to Better Represent the Clinical and Scientific Reality of ASD

**DOI:** 10.3390/ijerph22060816

**Published:** 2025-05-22

**Authors:** Mirah Dow, Ting Wang

**Affiliations:** School of Library and Information Management, Emporia State University, 1 Kellogg Circle, Emporia, KS 66801, USA; twang2@emporia.edu

**Keywords:** autism spectrum disorder, young children, DSM-5, neurodiversity, transdiagnosis

## Abstract

Objective: This study explores the current state of science regarding DSM-5 diagnostic criteria for Autism Spectrum Disorder (ASD) in young children. It examines the effectiveness of existing diagnostic methods and evaluates the potential of transdiagnostic approaches for early intervention. Method: A systematic literature review was conducted using MEDLINE, PsycINFO, and the Psychology and Behavioral Sciences Collection, focusing on peer-reviewed studies published between 2020 and 2023. The search followed PRISMA guidelines, selecting articles investigating ASD diagnosis in toddlers and preschoolers using DSM-5 criteria, behavioral assessments, and emerging diagnostic tools. Results: Findings indicate that DSM-5 provides a structured framework for ASD diagnosis, but it has limitations in early identification. It is necessary to integrate multiple assessment tools. Recent research highlights transdiagnostic models, which move beyond rigid diagnostic categories to capture the complexities of ASD presentation in young children. Conclusion: The literature supports a shift towards a transdiagnostic approach that combines behavioral, biological, and environmental assessments. This study underscores the need for interdisciplinary collaboration to refine ASD diagnostic frameworks to ensure more accurate and timely diagnoses that better serve affected children and their families.

## 1. Introduction

Few life circumstances cause parents more concern and stress than the realization that an infant’s developmental processes are uncharacteristic of that of a typically developing infant. Major questions arise when a young child in the early months of life experiences difficulties in basic human functioning such as feeding and digestion, sleeping, senses, reflexes, language development, and acquisition of specific communication and social skills. Over time, when questions are unanswered by healthcare and medical providers and the child’s needs escalate, everyday life, despite best efforts, quickly becomes intolerable for the child and overwhelming and intense for mothers as primary caregivers. As indicated in studies, a lack of specialized information results in tremendous uncertainty for families and caregivers and presents a public health crisis involving loss of quality of life and economic struggles associated with caring for a child at risk of various forms of neurodiversity [1,2,3,4]. This situation is particularly applicable to the many families and caregivers of a child with Autism Spectrum Disorder (ASD), one form of neurodiversity, a type of neurological difference that should be respected as one aspect of human differences.

Defined by the Diagnostic and Statistical Manual of Mental Disorders [5], ASD is a complex neurodevelopmental disorder beginning in early childhood through the lifespan. It is characterized by impairments in language development and communication, problem-solving, and social skills. Restrictive, repetitive behaviors and learning difficulties are common in ASD. According to the US Centers for Disease Control (CDC) and Prevention [6], in 2022, the prevalence of ASD is about 1 in 31 children aged 8 years (approximately 3.4 times more prevalent in boys (49.2) than girls (14.3)). While much dialog about ASD is focused only on a portion of the globe in the US, Canada, and Western Europe, according to Olusanya et al. [7] in a systematic analysis of the Global Burden of Disease Study 2016, “the global burden of developmental disabilities has not significantly improved since 1990, suggesting inadequate global attention on the developmental potential of children who survived childhood as a result of child survival programs, particularly in sub-Saran Africa and south Asia” (para. 4).

While some research indicates little support for intensive behavior interventions [8], according to McCarty and Frye [9], “[t]he only proven therapy for core symptoms of ASD is behavioral therapy, particularly if it is started early in life” (p. 1), and long-term outcomes for the child are more likely the earlier the intervention. They further indicate that “the mean age of obtaining a diagnosis of ASD is 4 years 3 months and has not improved significantly over the 2 past decades to educate the public and health professionals” (p. 2).

An ASD (APA, DSM-5, 2013) diagnosis, which first appeared as the Pervasive Developmental Disorder class of conditions in the DSM-3 [5], has been useful for some individuals who begin life experiencing neurodevelopmental difficulties. In the US, the DSM-5 is the ultimate diagnostic reference with target symptoms for ASD. However, the DSM-5 does not provide a formal test for ASD, nor for any mental health classification. DSM-5 criteria provide a framework for a medical diagnosis of ASD [10]. DSM-5 results are used to legitimize education, medical, and social systems support, and outcomes, with the lifetime social costs in the US estimated to be more than USD 7 trillion, which is predicted to increase to USD 11.5 trillion by 2029 if prevalence remains the same [11].

Without addressing ASD specifically and while identifying significant challenges for transdiagnostic science, Dalgeish et al. [12] raised major concerns about traditional forms of diagnosis, such as diagnoses based on the DSM-5. Although the DSM-5 presents a more updated classification system than the DSM-IV, Dalgleish et al. [12] argue that the traditional systems, which historically governed interventions and services, may no longer be sufficient. Instead, they advocate for a shift toward innovative approaches to conceptualizing and addressing mental health conditions, including alternative methods for understanding their onset and progression. Specifically, they emphasize, as does Jansen et al. [13], the potential benefits of a “transdiagnostic” (para. 1) approach, which moves away from rigid diagnostic categories toward a broader, integrative understanding of mental health problems. The perspective includes early interventions addressing functional difficulties, embracing neurodiversity as an accepted aspect of human variation, and developing new frameworks for responding to conditions such as ASD.

The current state of science about the DSM-5 diagnostic criteria for ASD and the future of proposed diagnostic innovations should be of high importance to parents, clinicians, and other stakeholders due to the widespread sociopolitical and pragmatic impact diagnostic tools have on children, their families, and/or caregivers, and society as a whole. When considering that distinct psychiatric diagnoses may not align with the complex configurations of human experience that manifest as ASD, it is prudent to also consider a biomedical model central to the current accepted diagnostic approach. Additionally, the confusion and anxiety experienced by parents and caregivers resulting in a frequent wait-and-see approach to diagnosis, putting diagnosis off indefinitely until the disorder is well established, may delay much-needed responses toward improving daily and lifespan quality of life. Given the high number of children presenting ASD characteristics in the US and in countries where information and resources are limited, de-stigmatizing children, mothers, and family members and providing opportunities and resources for children wherever located must be a worldwide priority.

### 1.1. Theoretical Framework

Dalgleish et al.’s [12] research and suggestions in favor of a rapidly developing transdiagnosis approach that cuts across traditional mental health diagnosis boundaries provide the theoretical background for this study. They address the theory of transdiagnosis and suggest that new “transdiagnostic approaches may have the potential to represent better the clinical and scientific reality of mental health problems” (para. 1), improving how researchers and medical practitioners think in various ways about the extremes of the human condition. Focused across the intellectual domains of classification and nosology, basic biopsychosocial research, and clinical science, they view transdiagnostic as a complicated and complex departure from traditional diagnosis that will cut across traditional diagnostic boundaries, present new classifications for mental health problems and new approaches to the management of various forms of diversity.

### 1.2. Problem, Research Question

To better understand the human condition through a major shift to transdiagnosis raises critical questions addressed in this literature review. What is the state of science about DSM-5 criteria and early diagnosis of ASD? What are the implications of the findings in current research for a proposed transdiagnostic approach to diagnosis and the steps to positively manage neurodiverse lives?

### 1.3. Systematic Search

The 2009 Preferred Reporting Items for Systematic Reviews and Meta-Analysis (PRISMA) Group guidelines were followed, which are frequently used in rigorous analysis of randomized trials and other types of medical interventions that must be precise [14]. The researchers also used the PRISMA-S checklist designed to verify that each component of the search is completely reported and reproducible. The search was carried out over the first half of 2024 in MEDLINE, PsycINFO, and Psychology and Behavioral Sciences Collection (EBSCO) to identify studies examining the consequences of applying the most current impairment criterion of DSM-5 [5] to infants and young children because they comprehensively cover medical, psychological, and behavioral research. The keywords “ASD” and “DSM-5” were used in conjunction with control terms “toddlers”, “preschoolers”, “and “early childhood” to conduct comprehensive searches in the three databases. Following pre-DSM-5 publication predictions, the search focused on articles published between 2020 and 2023 to capture the most recent research and developments in the field, ensuring the findings reflect the latest evidence and practices. The initial search yielded 308 articles, including 118 documents in MEDLINE, 167 in PsycINFO, and 23 in the Psychology and Behavioral Sciences Collection (EBSCO).

Identifying criteria for inclusion and exclusion of studies is essential for systematic review research protocols. Inclusion criteria outline the traits of the population that align with the research objective, and the exclusion criteria identify specific characteristics that, despite meeting the inclusion standards, are likely to introduce biases or compromise the integrity and reliability of the findings [15]. Articles included in the current study focus on young children, including toddlers and preschoolers, and explore the application and implications of DSM-5 diagnostic criteria in identifying ASD; peer-reviewed journal articles published between 2020 and 2023; and are published in English. While 2024 publications were reviewed, none applied to the present study. Articles excluded in the current study are those not relevant to the early diagnosis of ASD or were unrelated to the DSM-5 criteria; focus on teenager or adult populations or other developmental conditions unrelated to the diagnostic processes for ASD in toddlers and preschoolers; published before 2020; and written in other languages besides English.

After removing the duplicated 11 studies, 16 studies using languages other than English were excluded, following the established criteria and using the filtering options available in the databases, resulting in 281 studies (Figure 1). Articles published before 2020 were identified, and 194 articles were excluded, narrowing the selection to 87. By reviewing the titles and abstracts of the articles, 62 articles deemed unrelated to the study objectives were removed, leaving 26 articles. Finally, 11 articles without full-text access were excluded, resulting in a final dataset of 15 articles for analysis (Table 1). Among these, five articles were published in 2020, five in 2021, one in 2022, and four in 2023.

Among the 15 selected articles, diagnostic approaches varied based on the target population and research objectives. Some adopted relatively straightforward diagnostic methods tailored to their target population and research objectives. For instance, Harris et al. [16] and Coulter [17] employed DSM-5 criteria and the M-CHART to explore ASD subsets and symptom expression in young children. Some studies adopted a more focused approach to validate novel diagnostic methods. Hicks et al. [18], for instance, combined saliva microRNA detection as a non-invasive biomarker with DSM-5 criteria, providing an innovative and promising method for enhancing the accuracy of ASD diagnosis. Conversely, some clinical trials use multiple diagnostic tools to ensure a reliable assessment of ASD, such as Dow et al. [19], combining ADOS-T, MSEL, VABS-II, and parent-report questionnaires (ESAC), and video-recorded home observations, providing a detailed analysis of behavioral and environmental factors, improving diagnostic accuracy and yielding actionable insights for clinical practice. Some studies employed a systematic review approach to evaluate diagnostic methods for ASD and analyze broader research trends. For instance, Alrehaili et al. [20] and Pellecchia et al. [21] reviewed diagnostic modalities encompassing laboratory investigations, neuroimaging, medical history, and physical examination. The reviews provided an in-depth perspective on the diagnosis of ASD in young children, emphasizing the integration of behavioral, biological, and social science perspectives to enhance understanding of the complex disorders.
ijerph-22-00816-t001_Table 1Table 1Studies by article title, publication journal subject area, and findings from investigations of DSM-5 criteria on infants and young children (2020–2023).Article TitleSample SizeAgeDiagnostic ToolsJournal Subject Area Findings in Current ResearchMcCarty, P., and Frye, R. E. (2020, October). Early detection and diagnosis of autism spectrum disorder: Why is it so difficult? Seminars in Pediatric *Neurology*, *35*, 100831. [9] N/AN/AM-CHAT;ADOS;ADI-R;developmental screening;biomarkersPediatrics, NeurologyImproved training may increase the occurrence of practice by primary care physicians in ASD diagnosis.Kaba, D., and Soykan Aysev, A. (2020). Evaluation of Autism Spectrum Disorder in Early Childhood According to the DSM-5 Diagnostic Criteria. *Turk*
*Psikiyatri Dergisi*, *31*(2). [22]8018 to 72 monthsABC;CARS;AbBCPsychiatry and Mental HealthChildren under 7 are the riskiest DSM-5 group for lost diagnosis, which leads to progressive loss of functionality.Harris, H. K., Sideridis, G. D., Barbaresi, W. J., and Harstad, E. (2020). Pathogenic yield of genetic testing in autism spectrum disorder. *Pediatrics*, *146*(4). [16]50018 to 36 monthsGenetic test;DSM-5 criteriaPediatricsToddlers with DSM-5 ASD diagnosis should be recommended for genetic testing.Dow, D., Day, T. N., Kutta, T. J., Nottke, C., and Wetherby, A. M. (2020). Screening for autism spectrum disorder in a naturalistic home setting using the systematic observation of red flags (SORF) at 18–24 months. *Autism Research*, *13*(1), 122–133. [19]22818 to 24 monthsADOS-T;MSEL;VABS-II;video-recorded home observation; parent-report questionnaire;ESACNeurology, GeneticsASD screening tools are not accurate enough in routine screening of toddlers. SORF provides beneficial video-recorded samples of child and family.Hicks, S. D., Carpenter, R. L., Wagner, K. E., Pauley, R., Barros, M., Tierney-Aves, C., … and Middleton, F. A. (2020). Saliva microRNA differentiates children with autism from peers with typical and atypical development. *Journal of the American Academy of Child & Adolescent Psychiatry*, *59*(2), 296–308. [18]4332 to 6 years oldDSM-5 criteriaPsychiatry and Mental Health, Developmental and Educational Psychology Salivary microRNA is a non-invasive test that can improve accuracy in diagnosis of ASD in children.Harris, H. K., Lee, C., Sideridis, G. D., Barbaresi, W. J., and Harstad, E. (2021). Identifying subgroups of toddlers with DSM-5 autism spectrum disorder based on core symptoms. *Journal of Autism and Developmental Disorders*, 1–15. [16]50018 to 36 monthsDSM-5 criteriaDevelopmental and Educational PsychologySocial communication and restricted/repetitive behaviors may more precisely characterize subgroups within toddlers at time of ASD diagnosis.Kong, X. J., Sherman, H. T., Tian, R., Koh, M., Liu, S., Li, A. C., and Stone, W. S. (2021). Validation of rapid interactive screening test for autism in toddlers using autism diagnostic observation schedule™ second edition in children at high-risk for autism spectrum disorder. *Frontiers in psychiatry*, *12*, 737890. [23]3518 to 84 monthsRITA-T;ADOS-2Psychiatry and Mental HealthRapid Interactive screening Test for Autism in Toddlers (RITA-T) was found to be valid for screening toddlers at high risk of ASD, allowing initiation of services before formal diagnosis with DSM-5.Coulter, K. L., Barton, M. L., Robins, D. L., Stone, W. L., and Fein, D. A. (2021). DSM-5 symptom expression in toddlers. *Autism*, *25*(6), 1653–1665. [17]33612 to 36 monthsITC;M-CHATDSM-5 criteriaDevelopmental and Educational PsychologyContradicts earlier studies suggesting that restrictive and repetitive behavior may not be apparent until later in childhood.Pellecchia, M., Dickson, K. S., Vejnoska, S. F., and Stahmer, A. C. (2021). The autism spectrum: Diagnosis and epidemiology. In L. M. Glidden, L. Abbeduto, L. L. McIntyre, & M. J. Tassé (Eds.), *APA handbook of intellectual and developmental disabilities: Foundations* (pp. 207–237). American Psychological Association [21]N/AN/ADSM-5 criteriaDevelopmental and Educational PsychologyPresents ASD as one of seven conditions that result in intellectual and developmental disabilities. Addresses intellectual disabilities from multiple disciplines in biological, behavioral, and social science.Haffner, D. N., Bartram, L. R., Coury, D. L., Rice, C. E., Steingass, K. J., Moore-Clingenpeel, M., … and Group, N. E. D. (2021). The Autism Detection in Early Childhood Tool: Level 2 autism spectrum disorder screening in a NICU Follow-up program. *Infant Behavior and Development*, *65,* 101650. [24]6918–36 monthsDSM-5 criteriaADECCBCLDevelopmental and Educational PsychologyAutism Detection in Early Childhood is useful as a level 1 screening tool identifying children at risk for ASD in high-risk NICU.Khan, Z. U. N., Chand, P., Majid, H., Ahmed, S., Khan, A. H., Jamil, A., … and Jafri, L. (2022). Urinary metabolomics using gas chromatography-mass spectrometry: potential biomarkers for autism spectrum disorder. *BMC Neurology*, *22*(1), 101. [25]8524–84 monthsDSM-5 criteria biomarkerNeurologyUrine organic acid profiles are good discriminators between children with ASD and typically developing children.Alrehaili, R. A., ElKady, R. M., Alrehaili, J. A., and Alreefi, R. M. (2023). Exploring Early Childhood Autism Spectrum Disorders: A Comprehensive Review of Diagnostic Approaches in Young Children. *Cureus*, *15*(12). [20]N/AN/ADSM-5 criteriadiagnostic; instrument;medical history;physical examinations;laboratory investigation;neuroimaging;screening testMedicineVarious diagnostic modalities, including lab investigations and neuroimaging, contribute to early detection and comprehensive assessment of ASD.Saban-Bezalel, R., Avni, E., Ben-Itzchak, E., and Zachor, D. A. (2023). Relationship between Parental Concerns about Social–Emotional Reciprocity Deficits and Their Children’s Final ASD Diagnosis. *Children*, *10*(11), 1786. [26]8048–154 monthsDSM-5 criteriaPediatricsParental concerns about their child’s development regarding deficits in social-emotional reciprocity are significant in predicting a final diagnosis of ASD.Lavi, R., and Stokes, M. A. (2023). Reliability and validity of the Autism Screen for Kids and Youth. *Autism*, *27*(7), 1968–1982. [27]19748–222 monthsDSM-5 criteria;ASKYDevelopmental and Educational PsychologyWhen children outgrow early childhood, the Autism Screen for Kids and Youth with items related to DSM-5 criteria is a promising screening tool.Francés, L., Ruiz, A., Soler, C. V., Francés, J., Caules, J., Hervás, A., … and Quintero, J. (2023). Prevalence, comorbidities, and profiles of neurodevelopmental disorders according to the DSM-5-TR in children aged 6 years old in a European region. *Frontiers in Psychiatry*, *14*, 1260747. [28]2896 yearsWISC-V;CELF-5;Batería para la evaluación de los procesos de lectura revisada-prolece—r;TEDI-MATH;DCDQPsychiatry and Mental HealthNeurodevelopmental disorders (NDD) often coexist with other disorders, and it is rare for them to occur alone. There is evidence of presence of comorbidities in language disorders and ADHD. Low economic resources, lower levels of education of parents, and lifestyle habits that can be improved can alert clinicians to the possibility of NDD.


To further categorize and analyze the selected articles, the Scimago Journal and Country Rank (ScimagoJR) website was used to identify the subject areas of the journals in which the articles were published. ScimagoJR provides comprehensive information on the scope and impact of scientific journals, allowing researchers to determine the primary disciplines associated with each publication accurately Classifying the journals into their respective subject areas can contextualize the findings within specific fields, such as neurology, psychiatry, and psychology. The categorization facilitated a more nuanced understanding of the focus and implications of the studies, helping to highlight the inter-disciplinary nature of research on ASD diagnosis in young children and the possibility of transdiagnosis. Selected sources by global populations demonstrate rigor in methodology, including sample size, data collection, and analysis.

We organized studies in a carefully selected collection of post-publication DSM-5 studies. The research was conducted after the release of the criteria in 2013, from the four recent years (2020, 2021, 2022, and 2023) when there were publications that were representative of science at this time. No relevant publication was found in early 2024 at the time of the inquiry. A narrative review of selected studies includes the type of setting, population, design, measures, and broad findings and conclusions that highlight key points, main arguments, and controversies in publications. The new DSM-5 ASD criteria consolidated previously distinct diagnoses, such as Asperger’s Syndrome and Pervasive Developmental Disorder–Not Otherwise Specified, under the unified diagnosis of ASD [29]. The DSM-5 introduced two core symptom domains—deficits in social communication and interaction and restricted/repetitive behaviors or interests. The criteria also provide a nuanced framework for diagnosis based on severity levels and the need for support [29].

## 2. Results

Among the selected 15 articles, one is a handbook chapter published by the American Psychology Association, and the remaining 14 are academic journal articles published across 11 different journals. Notably, “Autism” and “Frontiers in Psychiatry” each contributed two articles (Table 2). The journals primarily focus on developmental and education psychology, psychiatry and mental health, and neurology. “Seminars in Pediatric Neurology”, “Autism Research”, and the “Journal of the American Academy of Child and Adolescent Psychiatry” cover more specialized areas, while “Pediatrics”, “Children”, and “Cureus” address broader fields like pediatrics and medicine.

The results section emphasizes exemplary methodological innovations, highlights emerging trends, and reflects shifting research priorities in diagnosing ASD. The selected studies are organized chronologically by publication year to capture the temporal evolution of scientific inquiry as it occurred, enhance clarity, and illustrate the progression of research as it developed. Figure 2 depicts the chronological evolution in ASD diagnostic approaches based on the highlighted studies’ methods and findings.

### 2.1. The 2020 Screening Procedures, Including the DSM-5 Standard

In recognition of the importance of early therapeutic interventions, the American Academy of Pediatrics recommends universal screening by pediatricians for ASD in infants 18 to 24 months using a variety of screening tools. To address the limitations of universal screening and the high rate of 1 in 3 children initially identified by screening tools for toddlers, McCarty and Frye [9] developed training for pediatricians to improve their use of screening tools toward the goal of reducing the number of toddlers without an ASD diagnosis referred for further evaluation and to decrease the time between parental concerns and time of diagnosis and care of children with ASD. In 2019, they examined the use of screening procedures, including the DSM-5 criterion, which is always documented when making a diagnosis at The Early Access for Care, Arizona Project at Phoenix Children’s Hospital, which began in 2015. They found several areas to improve efficiency and effectiveness, including unnecessary procedures that could result in wasted resources. For example, they discovered that the length of the Screening for Autism in Toddler and Young Children instrument designed for use in the physician’s office may explain why screening was not universally implemented and, further, that primary care physicians were not comfortable and did not document functional limitations found in children, which is necessary for the confirmation of ASD and obtaining services offered by the Arizona State Department of Developmental Disabilities. With the priority for identifying children early to initiate treatment, these researchers emphasized that diagnosis includes the efficient use of primary and secondary behavioral screening tools, in combination with the careful placement of five categories of biomarkers (prenatal, presymptomatic, diagnostic, sub-grouping, and treatment) in the diagnostic process (not in isolation) to stratify risk, diagnose, and assist with etiological classification and treatment.

### 2.2. The 2021 ASD Diagnosis Based on DSM-5 and Screening Tools

In consideration of significant ASD symptoms to necessitate the ASD diagnosis, Coulter et al. [17] addressed concerns about diagnosis using the DSM-5 given the occurrence of early atypical behaviors such as language delays, social communications deficits, and restricted and repetitive behaviors (RRBs) in young children not only presenting in ASD but also in children with other developmental disorders. They point out that there is agreement in the literature that social and sensory differences typically appear during the second year of life and that differentiating ASD from other developmental disorders is more difficult in children younger than 24 months. Consequently, they designed a study of toddlers initially screened during a pediatric visit using the Infant–Toddler Checklist (ITC) by Wetherby and Prizant [30]. Toddlers who raised ASD concerns were evaluated using the DSM-5 and other measures of cognitive abilities (Mullen Scales of Early Learning) by Mullen [31]; adaptive behaviors in communication, socialization, daily living, and motor skills (Vineland Adaptive Behavior Scales-2 (VABS-2)) by Sparrow, Cicchetti, and Balla [32]; play-based activities that show social effect and restricted and repetitive behaviors, and caregiver-report of ASD symptoms Toddler Autism Symptom Inventory (TASI)) by Coulter et al. [17]. Using all available data, the toddler participants (n = 294) were assigned to one of three diagnostic groups with a mean age of 21 months higher in the ASD group: ASD (71), other Developmental Disorder (131), and Typical Development/No Diagnosis (91).

Following a lengthy discussion of results in the Coulter et al. [17] study indicating differences among diagnostic groups, it was pointed out that caregivers of children 12–36 months recognize and report more social communication symptoms and repetitive behaviors when compared to children in developmental disorders or typical development/no diagnosis groups. The DSM-5 symptoms most recognized in the ASD group and found to be the most effective at discriminating among diagnostic groups were stereotyped/repetitive motor movements, object use, and speech, as well as deficits in relationships, deficits in social–emotional reciprocity, and restricted fixed interests. This contradicts earlier studies [33,34], suggesting that RRBs may not be apparent until later in childhood. A key conclusion positively supporting the use of the DSM-5 is that the TASI is useful through organizing operationalized toddler behaviors into DSM-5 symptom categories so that clinicians can determine symptoms that are not presented based on caregiver reports and can differentiate among diagnoses.

### 2.3. The 2022 ASD Diagnosis Advanced Based on Biomarker

A promising biomarker for a new perspective in addition to the clinical behavioral diagnosis of ASD was discovered by Khan et al. [25] in urine samples of 85 subjects in a case-control study in the Department of Pathology and Laboratory Medicine in collaboration with the Department of Pediatrics and Child Health, Aga Khan University, Pakistan. ASD subjects (n = 65) were diagnosed by a pediatric neurologist based on DSM-5 criteria and typically developing (TD) healthy controls from August 2019 to June 2021. Discriminant analysis showed that 3-hydroxyisovalericc, homovanillic acid, adipic acid, suberic acid, and indole acetic were significantly different between ASD and TD groups. Results showed that 88.2% of cases were correctly classified either as ASD or TD based on urine profiles.

### 2.4. The 2023 ASD Diagnosis Requires Behavior, Neurological, and Biological Measures

A comprehensive global review of early childhood ASD by Alrehaili et al. [20] indicates that ASD is a cluster of developmental disabilities that align with the diagnostic criteria in the DSM-5. They acknowledge studies that have demonstrated that ASD “can be identified at or below 18 months of age, with a more accurate evaluation possible by age 2” (p. 6). Without discussion of the merits of DSM-5 diagnostic criteria in diagnosing ASD, they assert that diagnosis of ASD in early childhood laying a roadmap for treatment requires a comprehensive, systematic, and structured approach through “behavioral observations, clinical presentations, or various imaging techniques such as functional MRI and diffusion imaging” [20], p. 1, and they stated that the objective of ASD evaluation is to determine effective care for the child in the context of the family. These researchers further state that evaluation should go beyond behavioral assessments to identify signs of illness, self-harm, and potential ill-treatment, and evaluation for ASD should include “neurological assessment, dysmorphism regulation, scrutiny for neurocutaneous stigma, and Woods light (ultraviolet light) examination” [20], p. 1. They conclude that screening tools such as the Communication and Symbolic Behavior Scales; the Infant-Toddler Checklist; Ages and Stages Questionnaires; Modified Checklist for Autism in Toddlers; and the Parents’ Evaluation of Developmental Status have only “a pilot role in the early detection of ASD” [20], p. 7, and ASD diagnosis should include lab investigations of potential biomarkers such as elevated whole-blood serotonin, melatonin deficits, identification of small-molecule peptides, proteins/peptides in cerebrospinal fluid, and expression of apolipoproteins in serum, and neuroimaging revealing insights about neuroanatomical structures and emotional processing.

In the DSM-5, ASD is considered a mental health problem and has, therefore, been addressed through the lens of the psychiatric diagnostic paradigm, likely influenced by general medicine and beliefs about states of health and causes of physical illnesses and diseases. The DSM-5 presents a diagnostic taxonomy useful for communication across stakeholder groups and facilitating clinical assessments and interventions. ASD symptoms are considered present or absent. Young children may appear to have symptoms that satisfy more than one developmental disorder, and children with the same diagnosis can have different symptoms. The DSM-5 criteria raise questions about common symptoms in typically developed young children, such as a strong desire for sameness in everyday life routines and fixed interests such as specific activities, objects, or subjects. ASD symptoms can change across developmental levels and the life span. These challenges outlined in Figure 2 are evidenced in the following studies presented as examples of research addressing diagnostic criteria.

## 3. Discussion

During the past two decades, studies, particularly in recent years published in the US, have progressively become more specific about the ineffectiveness of diagnostic criteria for ASD in the DSM-5 and more detailed as to what behavioral screening tools and laboratory tests should be included in the diagnostic process for young children. This is particularly momentous given that for approximately 40 years, the Diagnostic and Statistical Manual of Mental Disorders published by the American Psychological Association has been the goal standard in the US and other regions of the world for the diagnosis of what is known today as ASD.

Early writing of the DSM-5 was carefully reviewed before the 2013 publication for opportunities to improve the manual with revisions, including making the new manual a more useful diagnostic tool in primary care settings, resulting in more diagnoses and establishing a more comparable international statistical classification of mental disorders [35]. Even before the 2013 publication, there were controversies, concerns, and researchers’ questions about the appropriateness of the DSM-5 criteria for toddlers at risk of developmental disorders [36]. For example, in an extensive study of toddlers (n = 2721), Matson et al. [37] found that fewer toddlers were diagnosed with ASD when DSM-5 criteria were applied. They projected—although the projection was unfounded—that the DSM-5 would result in far fewer persons being diagnosed and qualifying for needed autism services. Extending findings of previous studies, Barton et al. [33] investigated the sensitivity and specificity of proposed DSM-5 criteria for ASD in toddlers who were believed to be vulnerable in the proposed DSM-5 when compared to the DSM-IV-TR in terms of stringent requirements of impairment in all three social-communication symptoms and two restricted/repetitive symptoms. Their sample from the coastal, eastern US was 422 toddlers aged 16.79–39.36 months who had screened positive on the Modified Checklist for Autism in Toddlers–Revised. The Autism Diagnostic Interview^®^–Revised (ADI^®^-R), the Autism Diagnostic Observation Schedule (ADOS), and DSM-5 evaluations were administered and data analyzed using SPSS 18.0. Receiver operating characteristic curves were generated using the sum of the ADI-R and ADOS threshold items contributing to each DSM-5 criterion. More participants with ASD were identified than in earlier studies in relatively high sensitivity and low specificity.

According to these researchers, “[s]ymptom presentation in young children is often less clear as symptoms may still emerge when children are referred for early evaluations. Toddlers’ behavior may also be more affected by situational variables. Their parents may have less experience than the parents of older children with developmental processes and age-related expectations” [33], p. 1190. It was noted that “about half of the children who did not have ASD were also identified using these cutoffs, suggesting that a mild endorsement of one aspect of a DSM-5 criterion may not meet the threshold for clinically significant impairment characteristics of autism” [33], p. 1190. The researchers concluded that the proposed DSM-5 should be modified because it may be too stringent for children younger than 3 years old.

Widely recognized databases for the dissemination of peer-reviewed publications searched in this review of scientific studies reveal documented, long-standing concerns about DSM-5 criteria for ASD. This documentation indicates deeply held concerns and repeated, significant efforts to justify DSM-5 diagnostic criteria in identifying young children with ASD. DSM-5 validation efforts involved the use of various screening tools wherein gathered observation and interview data are statistically evaluated for alignment with DSM-5 criteria. The greater the degree of alignment found in the study, the stronger a case for believability may be made for significance in determining an ASD diagnosis. In contrast, there is strong evidence that suggests that behavioral assessments and diagnostic frameworks, including screening tools and the DSM-5 criteria, are insufficient without laboratory investigation of potential biomarkers as measurable indicators of biological states or conditions in distinguishing ASD from typical development.

Recent studies addressing DSM-5 diagnostic criteria have in common that they focus on practices of using various screening tools in combination with the DSM-5, which alone is frequently insufficient in diagnosing young children. Albeit a quest to improve quality of life, researchers’ determination to develop and use screening tools designed to gather observation data from clinicians, families, caregivers, and biomedical data may be partly motivated by several factors identified in the present literature review. First, the suspected prevalence of ASD is increasing at a staggering rate each year. There is wide agreement that in young children, ASD can be differentiated from other developmental disabilities, and behavioral interventions are most effective early in the life of a child [38,39,40]. There are life-long, high personal and societal financial costs directly associated with ASD. Despite the compelling recommendation by the American Academy of Pediatrics that children be screened for ASD during regular well-child doctor visits at 18 and 24 months, and despite organized educational efforts to improve universal screening by PDPs, there is evidence that many PCPs in the US are reluctant to conduct screening and/or to document limitations. With early identification of ASD as an integral function of the primary care medical home and the responsibility of pediatric healthcare providers, there appears to be a huge gap—a crisis—in identifying and diagnosing young children who present symptoms of ASD, potentially leaving many young children and families without frontline assistance and care.

Given the unintended yet complicated and complex state of science dominated by DSM-5 diagnostic criteria for ASD and the many young children experiencing distress in need of specialized care, a new theoretical way of thinking, as suggested by Dalgleish et al. [12], or a paradigm shift in the current state of science, is needed that has the potential to reveal interpretations of neurologically diverse development and supports beginning in early childhood. Recent research indicates that machine learning analysis of motor patterns and eye movements can accurately identify ASD with up to 93% accuracy, offering the potential for faster, more objective screening tools to supplement traditional behavioral assessments [41,42,43,44]. Rather than dispense with the current diagnostic system altogether, it appears logical to adopt what Dalgleish et al. [12] describe as a “soft trans-diagnostic” (p. 182) approach, keeping the “underlying diagnostic classification while seeking to elucidate processes or develop interventions that have relevance to one or more of the diagnoses as traditionally formulated” (p. 182).

### 3.1. New Soft Transdiagnostic Model

The current study suggests possibilities for an Integrated Transdiagnostic Approach Model (ITAM) that combines behavioral, biological, and environmental assessments to understand the clinical and interpretive scientific realities of ASD among young children (Figure 3). The ITAM incorporates existing diagnostic tools while integrating multi-modal evaluation to ensure more accurate, individualized, and timely diagnoses. Consistent with the position of Dalgleish et al. [12] that a transdiagnostic approach emphasizes functional impairments over rigid diagnostic labels, the new ITAM integrates three core components: behavioral screening, biological testing, and environmental observation. Behavioral assessments, such as the Modified Checklist for Autism in Toddlers (M-CHAT) and Autism Diagnostic Observation, Second Edition (ADOS^®^-2), provide a foundation for identifying social and communicative deficits. However, the ITAM expands the behavioral evaluation by incorporating biomarkers, including saliva microRNA profiles, urinary metabolomics, and genetic testing, as highlighted in recent studies [18,25]. In addition, environmental factors, such as caregiver-reported concerns, video-recorded home observations, and socio-economic context, are integrated into the diagnostic process to acknowledge external roles in child development.

A tiered screening approach is recommended to operationalize the ITAM. During routine well-child visits, pediatricians conduct primary screening using tools like the Rapid Interactive Screening Test for Autism in Toddlers (RITA-T) or the Modified Checklist for Autism in Toddlers, Revised (M-CHAT-R/F). If concerns are identified, it should be considered that children undergo a comprehensive diagnostic evaluation, which may include standardized tools such as the Autism Diagnostic Observation Schedule, Second Edition (ADOS^®^-2), alongside clinical assessments based on DSM-5 diagnostic criteria and consideration of biological and environmental evaluations. While biological evaluations, such as genetic testing or biomarker research, have shown emerging promise in research contexts, they are not yet established for routine clinical diagnosis due to the complex and multifactorial etiology of ASD. The ITAM emphasizes the collaborative care pathway to ensure the findings are shared with a multidisciplinary team, such as parents and caregivers, pediatricians, psychologists, geneticists, and educators, to facilitate individualized intervention plans. With ongoing monitoring and periodic re-evaluation, tracking developmental progress, and adjusting interventions as needed, the use of the ITAM has the potential to accurately and effectively represent realities experienced by the child. The ITAM has the potential to improve diagnostic accuracy by combining behavioral, biological, and environmental indicators, reduce false positives and missed diagnoses, and promote early intervention. While the ITAM will require validation through clinical trials, it presents a promising pathway toward inclusive, precise, and effective ASD diagnosis and care.

### 3.2. Future Meta-Theoretical Assumptions

For there to be substantive change, there must be the desire to adopt a different set of meta-theoretical assumptions about science that address ASD not as a mental health condition but as research topics focused on ASD as a lifelong neurodevelopmental condition. In keeping with Kuhn’s [45] notion of revolutionary science, for researchers and theorists to switch paradigms calls for a change, which is possible but not often achieved in practice. Burrell and Morgan [46] offer four paradigms for understanding the nature of science and society “in the hope that knowledge of the competing points of view will at least make us aware of the boundaries within which we approach our subject” (p. 25). Table 3 highlights two of Burrell and Morgan’s [46] four conceptualizations of social science paradigms with brief descriptions of each. This table describes the nature of each paradigm and shows clear boundaries between two paradigms that are relevant to the current state of science about ASD, wherein the functionalist paradigm is dominant, and the interpretivism paradigm appears to be emerging. Each paradigm, or lens through which ASD is viewed, has its accepted theory, assumptions, models, practices, and tools.

The functionalist paradigm has provided the dominant framework for the conduct of academic psychology and the study of the mind and behavior. The interpretive paradigm is concerned with understanding the social construction of reality, the way people create and share meanings. Awareness of the boundaries of these two social science paradigms can provide some signposts in a roadmap for conversations about a shift in the science of ASD to the transdiagnosis paradigm and guide an accelerated shift to more accepted theories, assumptions, models, practices, and tools useful in the diagnosis of ASD in the earliest stages of life and through the life span.

A strong case for an integrated transdiagnostic approach emerges from the limitations of the functionalist paradigm. Traditional diagnostic methods grounded in the functionalist paradigm primarily focus on observable behavioral symptoms, often missing underlying neurobiological variability and the complex, subjective experiences of individuals with ASD. A proposed integrated transdiagnostic model offers a more holistic understanding, drawing insights from multiple disciplines such as genetics, neurobiology, developmental psychology, and lived experience narratives. The integration has the potential to improve diagnostic sensitivity, particularly in early childhood and among diverse populations, and better inform personalized intervention strategies. Research into overlapping biomarkers and the recognition of dimensional traits that cross diagnostic categories (such as sensory sensitivities across neurodevelopmental conditions) supports the need for a transdiagnostic perspective [25]. Although further empirical validation is needed, this approach is promising for advancing precision medicine and inclusive care.

The literature on early diagnosis of ASD has been published in the early years of the current decade in fields such as developmental and educational psychology, psychiatry, and neurology, demonstrating the diversity and interdisciplinary nature of research on ASD and the need for input from different scientific disciplines. However, the limited disciplines may also indicate the inadequacy of transdiagnostic approaches. While studies have demonstrated the potential of biomarkers for diagnosing ASD [25], more extensive efforts, such as promoting collaboration between researchers from various disciplines to create a more comprehensive understanding of ASD, are necessary to fully recognize and implement the benefits of biomarker-based diagnosis in clinical practice and to scale up transdiagnosis. Expanding research in this area may lead to more precise diagnoses, enabling personalized interventions and improving treatment outcomes for individuals with ASD [1,3].

### 3.3. Future Research

Research must acknowledge, uphold, and build on the many recent contributions of science to understanding ASD, as well as the existing high standards and expectations for the diagnosis of young children with ASD. Fortunately, licensed clinical psychologists, psychiatrists, and physicians are accountable through their formal education and oath, pledging to act for the patient’s good [47]. Because of the nature of their job and their moral and ethical responsibilities, making a diagnosis of ASD is high stakes for them, as well as for a young child and their parents and caregivers. Without a medical diagnosis, communicating a young child’s strengths and challenges and informing interventions are necessary, and community support is frequently out of reach. Without minimizing professional responsibilities or progress in today’s society, a transdiagnosis approach such as the proposed soft ITAM to the diagnosis of ASD has the potential to widen the functionalist lens, enabling an interpretive focus by researchers beyond the current occupation with DSM-5 criteria, allowing future research questions such as the following: What are individual, economic, educational, medical, and social service benefits of access, dissemination, and use of research findings about ASD diagnostic criteria through open access journals? How can the involvement of parents and caregivers of young children who present symptoms of ASD positively improve the training of primary care physicians? How can adults with high literacy skills, minimal support, and those who personally experience neurodiversity contribute to future discussions about DSM criteria for ASD?

In addition, future studies can explore interdisciplinary approaches to enhance the accuracy, speed, and accessibility of ASD diagnosis. While ASD diagnosis by physicians or health professionals using the DSM-5 is the current dominant standard, incorporating new scientific findings from genetics, metabolomics, and proteomics will provide complementary insights and improve understanding of ASD. Fostering interdisciplinary research that combines insights from various disciplines, such as genetics, metabolomics, and proteomics, may provide possibilities to improve diagnostic practice and a comprehensive understanding of ASD.

### 3.4. Study Limitations

Despite the researchers’ best efforts, the study has some limitations. The study used only three databases—MEDLINE, PsycINFO, and the Psychology and Behavioral Sciences Collection (EBSCO). The databases are comprehensive; however, it is possible that some relevant studies in other databases were not included in the current literature review, which may limit the review scope. In addition, the searches used specific keywords and control terms, such as “ASD”, “DSM-5”, and “toddler, preschoolers, and early childhood”, limiting the breadth of the search. While these terms are relevant, other relevant terms, or variations, could generate additional studies.

## 4. Conclusions

ASD is widely understood as a complex neurodevelopmental disorder that appears in the early months of life. Experience discloses that there are many children thought to present characteristics of ASD, many of whom likely do not have access to assessment by licensed psychologists, psychiatrists, or physicians. Initial medical evaluations should be made by medical “professionals who are experienced and fully qualified to make this diagnosis” [48], para. 1. The dominant criteria established for confirmation of ASD are outlined in the DSM-5 [11,49]. A focused review of recent scientific literature reveals the widely held current state of science about DSM-5 criteria and early diagnosis of ASD. Existing literature is replete with difficulties in diagnosis primarily associated with alignment to DSM-5 criteria for ASD that are recognized and significantly addressed by researchers, clinicians, and practitioners. There is growing recognition that developing and using biomarkers and MRIs in conjunction with screening methods such as the DSM-5 and a variety of others improves the diagnostic process and, subsequently, has the potential to enhance the quality of life and the prognosis for young children who begin life with symptoms uncharacteristic of a typically developing child. Studies indicate that frontline screening practices by primary care physicians are insufficient in addressing the urgent needs of young children and their families. Studies implicate urgency for a shift in the dominant diagnosis paradigm and a new lens for viewing ASD as an accepted, lifelong form of neurological development. The transdiagnosis lens and approaches such as the ITAM suggested in this study, when accepted for designing research and communication, will open doors to new approaches to scientific discoveries about ASD as a neurodiverse, lived experience and provide interdisciplinary theories, models, practices, and tools with the potential for improving care, and legitimize a continuum of supports during early childhood and on the frontlines of a lifetime. 

## Figures and Tables

**Figure 1 ijerph-22-00816-f001:**
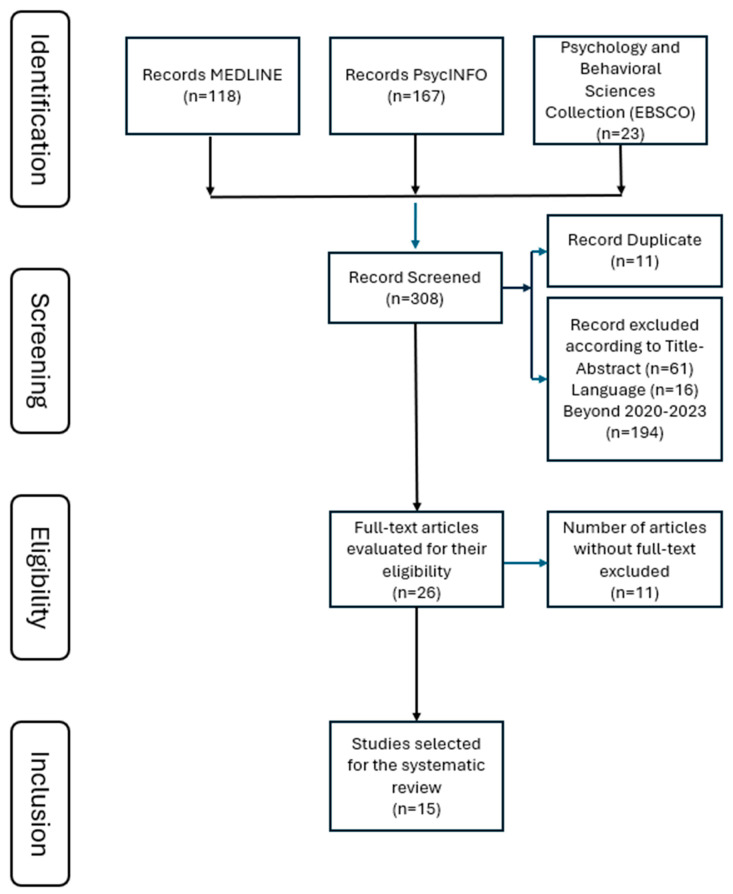
PRISMA flow diagram. Note: Adapted from PRISMA 2009 Flow Diagram [14]. Copyright: © 2009 Moher et al. This is an open-access article distributed under the terms of the Creative Commons Attribution Licence, which permits unrestricted use, distribution, and reproduction in any medium, provided the original author and source are credited.

**Figure 2 ijerph-22-00816-f002:**
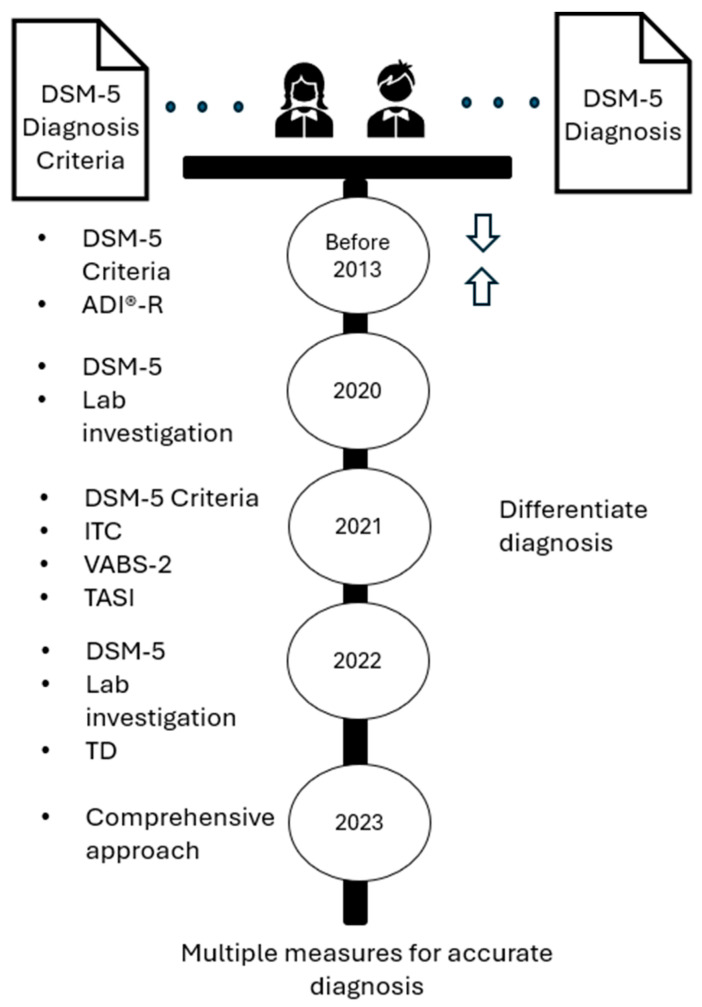
Progression of identified controversies about diagnostic appropriateness of DSM-5.

**Figure 3 ijerph-22-00816-f003:**
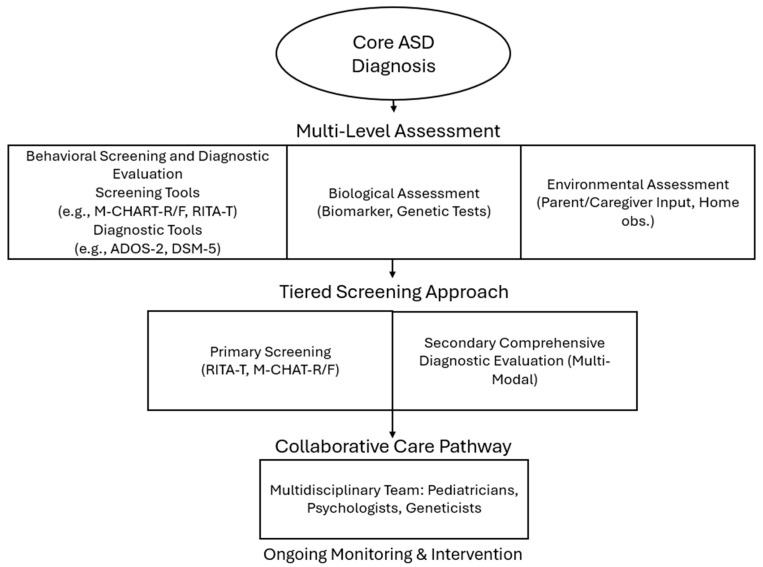
Integrated transdiagnostic approach model.

**Table 2 ijerph-22-00816-t002:** Studies by journal title, journal frequencies, and journal subject area.

Journal Title	Journal Frequencies	Journal Subject Area
Autism	2	Developmental and Educational Psychology
Frontiers in Psychiatry	2	Psychiatry and Mental Health
Seminars in Pediatric Neurology	1	Pediatrics, Neurology
Turk Psikiyatri Dergisi	1	Psychiatry and Mental Health
Pediatrics	1	Pediatrics
Autism Research	1	Neurology, Genetics
Journal of the American Academy of Child and Adolescent Psychiatry	1	Psychiatry and Mental Health, Developmental and Educational Psychology
Journal of Autism and Developmental Disorders	1	Developmental and Educational Psychology
Infant Behavior and Development	1	Developmental and Educational Psychology
BMC Neurology	1	Neurology
Children	1	Pediatrics
Cureus	1	Medicine

**Table 3 ijerph-22-00816-t003:** Characteristics of paradigms of social science thought to identify and guide distinct intellectual traditions.

Dominate Paradigm—Diagnostic Paradigm for ASD The Functionalist Paradigm [20]	New Paradigm—Transdiagnosis Paradigm for ASD The Interpretivist Paradigm [20]
Dominate framework (Western world) Grounded in the sociology of regulation, a microscopic view of social reality and objective point-of-view Seeks rational explanations of social affairs for prediction and control, approach to science is premised on logical positivism Micro-objective Apply positivistic methodologies to microscopic social phenomena, such as patterns of behavior, action, and interaction, in an attempt to predict and control social life Argues that all events and structures in society are functional because if they were not, they would not exist Belief that current aspects of society are indispensable to the system and, as such, all structures that exist should continue to exist	Emerging framework (Western world) Implicitly committed to regulation and order Assume that the social world is cohesive, orderly, and integrated, they (unlike the functionalists) are oriented toward understanding the ongoing processes through which humans subjectively construct their social worlds Addresses the same social issues as the functionalist paradigm but concerned with understanding the essence of the everyday world as an emergent social process When a social world outside the consciousness of the individual is recognized, it is regarded as a network of assumptions and inter-subjectively shared meanings. Micro-subjective Concerned with understanding the social construction of reality—the way people create and share meaning

## Data Availability

This study is a literature review and does not involve the collection or analysis of primary data. The findings reported in this manuscript are derived from the synthesis of previously published research articles, reports, and publicly available sources. All data sources are cited appropriately within the manuscript. No new empirical data were generated or used in the development of this review. This research is submitted now to only the Journal of Consulting and Clinical Psychology and has not been published in any journal.

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
