# Peer review of "Health Knowledge About Early Diagnosis of Autism Spectrum Disorders: A Case for Soft Transdiagnostic Approaches to Better Represent the Clinical and Scientific Reality of ASD"

_ijerph, 2025, doi:10.3390/ijerph22060816_

Round 1

Reviewer 1 Report

Comments and Suggestions for Authors

The manuscript analyzes approaches to the diagnosis of ASD in children, mainly of young age (up to 3-4 years old). It is known that the accuracy of ASD diagnosis using traditional approaches at an early age is questionable. This determines the particular relevance of this study.

The empiricalbasis of the articleis the results of a systematicreview of publicationsonthistopicfrom2020to2023.Such a narrowtimeperiod is justified in order toreflect the latestdataandadvancesinthefield.

The mostimportantresults of the study are summarized as follows. The DSM-5criteriaprovide a productivebasisforASD earlydiagnosis,butscreenings application based on the DSM-5criteria is notsufficientforveryyoungchildren. The authorssuggest that diagnosticaccuracycanberadicallyimprovedbyapplying a "trans-diagnosticapproach."Thisapproachimplies the application of interdisciplinaryresearchresultsindiagnostics.Theseresults make it possible to complementtraditionaltoolsbasedon the DSM-5criteria with innovativeapproachestoASDdiagnosis,suchas the inclusion of salivamicroRNAprofiles,urinebiomarkers,genetictesting, and homesurveillancevideos.

The manuscriptisclear,relevant to the topicandwellstructured. The empiricalbasis of the studyis the content of a seeminglyinsignificantnumber of articles,only15.Butsuch a smallnumber is justified,firstly, by the narrowrangeof the specifiedpublicationtime(2020-2023), and secondly, by fairlyclearandreasonablecriteria for findingsources.

The comments are related to typos:

Line431-AITMinstead of ITAM.

Line 447 is a reference to Table 1, in fact it is Table 3.

 Line 474 - don't forget to replace the links!

Author Response

Thank you very much for the carefully review!

Line431-AITM instead of ITAM.

We made the change. 

Line 447 is a reference to Table 1, in fact it is Table 3.

We made the change. 

Line 474 - don't forget to replace the links!

We made the change. 

Reviewer 2 Report

Comments and Suggestions for Authors

I am pleased to have had the opportunity to review the article titled "Health Knowledge about Early Diagnosis of Autism Spectrum Disorders: A Case for Soft Transdiagnostic Approaches to Better Represent the Clinical and Scientific Reality of ASD." I find this topic to be of particular relevance within the clinical community, and it is essential that it receives due attention from the scientific community as well.

The introduction is excellently crafted. The authors skillfully emphasize the challenges faced by traditional diagnostic processes, especially due to the significant variability in ASD symptomatology.I also appreciate the discussion on how the clinical approach to diagnosing ASD has evolved over the years.

1.However, I believe the results section could be improved with some refinements. Clarifying the rationale behind organizing the results by years, as this is the authors' chosen approach for presenting their findings, but it is not the typical way readers expect to see the results presented. Introducing this aspect earlier could help set clearer expectations for the reader. I suggest providing a more detailed explanation before formally presenting the subsections (2.1, 2.2, 2.3, 2.4).

2.Additionally, I suggest to include a summary table for these four subsections. It could further enhance clarity and effectively highlight the key points the authors wish to emphasize.

3.I also noticed that there is limited discussion on non-invasive biomarkers, such as wearable sensors or motion analysis, which is an increasingly debated and relevant topic in this field. I recommend expanding the discussion to include these aspects, as they could provide valuable insights into early ASD diagnosis. Below are some references that might be helpful:

  • Simeoli, R., Rega, A., Cerasuolo, M., Nappo, R., & Marocco, D. (2024). Using Machine Learning for Motion Analysis to Early Detect Autism Spectrum Disorder: A Systematic Review. Review Journal of Autism and Developmental Disorders. https://doi.org/10.1007/s40489-024-00435-4
  • Luongo M, Simeoli R, Marocco D, Milano N, Ponticorvo M. (2024) Enhancing early autism diagnosis through machine learning: Exploring raw motion data for classification. PLoS One. 22;19(4):e0302238. doi: 10.1371/journal.pone.0302238. PMID: 38648209; PMCID: PMC11034672.
  •  Anzulewicz, A., Sobota, K., & Delafield-Butt, J. T. (2016). Toward the autism motor signature: Gesture patterns during smart tablet gameplay identify children with autism. Scientific Reports, 6, 1–13. https://doi.org/10.1038/srep31107
  • Lin, Y., Gu, Y., Xu, Y., Hou, S., Ding, R., & Ni, S. (2022). Autistic spectrum traits detection and early screening A machine learning based eye movement study. Journal of child and adolescent psychiatric nursing official publication of the Association of Child and Adolescent Psychiatric Nurses Inc, 35(1), 83–92. https:// doi.org/10.1111/jcap.12346 

4.Finally, I would like to point out a typographical error on line 377.

Author Response

Thank you very much for the constructive comments! They are very helpful. 

1. I believe the results section could be improved with some refinements. Clarifying the rationale behind organizing the results by years, as this is the authors' chosen approach for presenting their findings, but it is not the typical way readers expect to see the results presented. Introducing this aspect earlier could help set clearer expectations for the reader. I suggest providing a more detailed explanation before formally presenting the subsections (2.1, 2.2, 2.3, 2.4).

Response: We addressed the rational for organizing the results by year before section 2. 

2.Additionally, I suggest to include a summary table for these four subsections. It could further enhance clarity and effectively highlight the key points the authors wish to emphasize.

Response: We explained that Figure 2 is a summary of the four subsections.

3.I also noticed that there is limited discussion on non-invasive biomarkers, such as wearable sensors or motion analysis, which is an increasingly debated and relevant topic in this field. I recommend expanding the discussion to include these aspects, as they could provide valuable insights into early ASD diagnosis. Below are some references that might be helpful:

  • Simeoli, R., Rega, A., Cerasuolo, M., Nappo, R., & Marocco, D. (2024). Using Machine Learning for Motion Analysis to Early Detect Autism Spectrum Disorder: A Systematic Review. Review Journal of Autism and Developmental Disorders. https://doi.org/10.1007/s40489-024-00435-4
  • Luongo M, Simeoli R, Marocco D, Milano N, Ponticorvo M. (2024) Enhancing early autism diagnosis through machine learning: Exploring raw motion data for classification. PLoS One. 22;19(4):e0302238. doi: 10.1371/journal.pone.0302238. PMID: 38648209; PMCID: PMC11034672.
  •  Anzulewicz, A., Sobota, K., & Delafield-Butt, J. T. (2016). Toward the autism motor signature: Gesture patterns during smart tablet gameplay identify children with autism. Scientific Reports, 6, 1–13. https://doi.org/10.1038/srep31107
  • Lin, Y., Gu, Y., Xu, Y., Hou, S., Ding, R., & Ni, S. (2022). Autistic spectrum traits detection and early screening A machine learning based eye movement study. Journal of child and adolescent psychiatric nursing official publication of the Association of Child and Adolescent Psychiatric Nurses Inc, 35(1), 83–92. https:// doi.org/10.1111/jcap.12346 

Response: We included the suggested articles in the manuscript. 

4.Finally, I would like to point out a typographical error on line 377.

Response: We do not see the typing error. 

Reviewer 3 Report

Comments and Suggestions for Authors

This was a systematic review using PRISMA-S checklist, however much the article reads more like a theoretical article on changing the diagnostic paradigm. There is a table summarizing findings from the 15 articles that met criteria however the consolidation of the findings from these studies is does not appear in the results section (for example, the information starting at 153 starting with “Among the 15 articles diagnostic approaches …” would make more sense in the results section). Lastly the construct of intergraded transdiagnostic approach is not well explained nor is the connection between the results of the review explained well. I also got the feeling the purpose of the study was not to do a systematic review but to support a conclusion you had already come to – transdiagnostic diagnostic paradigm.

General concept comments
Article: While the article contains some interesting ideas the second half of the article does not read like a systematic review. The description of the literature search is understandable and appeared to follow standard search practice however, I expected more discussion of what was found in the literature search.
Review: Overall, I found this article in need of considerable work to bring it up to the quality standards of a systematic review. It also felt like it was two articles one more about a new paradigm and one a review article. As a review article there needed to be more synthesis of the results of the review. Some of the constructs in this article are interesting so you might want to do an extensive re-write. If this is the case, I encourage you to clarify the data around diagnostic assessments. For example, the age of an ASD diagnosis is now under 4 years in the US and in some states (e.g. California) it is even earlier. See reference to latest CDC report. This report just came out so you wouldn’t have had this information when writing the article. Additionally, there is data showing that children diagnosed early may no longer meet the criteria for ASD when they reach school age. (Blumberg et al., 2016; Harstad et al., 2023). Lately, I found many sentences overly long making it harder for the reader to follow I suggest looking at sentience length. 

Specific comments: Please see my specific comments below.

  1. 39: Define neurodiversity – might not even need to use the neurodiversity construct. The article does not expand on the term later so I wouldn’t use it. If neurodiversity is important to the contract of a new diagnostic paradigm I would expand.
  2. 47: New prevalence rates just came out – update rates are 1 in 31 in the US.
  3. 56: more recent studies question the need for intensive behavioral treatment (high hours) for young children (See Sandbank et al., 2024). I understand the point you are trying to make is that early diagnosis is important I think you can make that point without the McCarty and Frye quote.
  4. 59: update average of diagnosis with new CDC report. It has gone down.
  5. 66: I don’t believe the DSM does provide formal assessment tools for any disorder. This statement implies that ASD is different in that regard but that is not the case.
  6. 79: It’s not clear what transdiagnostic diagnosis really is. This is where more information about what your mean my neurodiversity could include if you decide to go in that direction. I the connection between the review of the 15 articles and transdiagnostic diagnosis was a little unclear.
  7. 335: Check updated screening data.
  8. 339: Matson et al. (2012) projection that there would be fewer children diagnosed did not occur. More children have been diagnosed. Latest CDC report increasing trend of diagnosis for young children. Thus, this point doesn’t strengthen your argument.
  9. 361: Prevalence rates and median age of diagnosis in California is 36 months, Texas is higher (see CDC report). The age range of diagnosis by state weakens this statement, because it suggests that it is not the DSM-5 criteria but differences in access to assessment by state.
  10. 371: DSM-5 should not be categorized as “behavioral assessment.”
  11. 381: Is there really wide agreement on the point of differentiation? There needs to be a citation here. I am not aware of this agreement or data that shows it is easy to differentiate.
  12. 423: The proposed secondary assessment mentions the M-CHAT or ADOS. The M-CHAT is a screening tool (often used by pediatricians) and not in the same diagnostic category as the ADOS which is not a screening tool. Thus, this suggestion may not be something diagnosticians would do. It is also not clear what is meant by a DSM-5 based assessment?
  13. 435: The suggestions for biological assessment (biomarker) does not have enough support. Only one study was cited that is related to a biomarker. I don’t think we have enough data to say there is biomarker. The etiology of autism is complex.
  14. 439: While I appreciate the intention to improve diagnostic paradigm for autism, I don’t think a strong case was made for an intergraded transdiagnostic approach.
  15. 498: Many diagnostics assessments for ASD are conducted by psychologists
  16. 495: Machine learning and AI kind come out of the blue in the later part of the paper. I would suggest that if you are going to bring these up that you reference them earlier.
  17. 517: A medical diagnosis can be made by non-physicians and this is done everyday

References that may be useful is you decide to rewrite the article.

Blumberg, S. J., Zablotsky, B., Avila, R. M., Colpe, L. J., Pringle, B. A., & Kogan, M. D. (2016). Diagnosis lost: Differences between children who had and who currently have an autism spectrum disorder diagnosis. Autism : the international journal of research and practice20(7), 783–795. https://doi.org/10.1177/1362361315607724

CDC report:

Shaw et al., 2025

https://www.cdc.gov/mmwr/indss_2025.html?ACSTrackingID=USCDC_921-DM146435&ACSTrackingLabel=MMWR%2520Surveillance%2520Summaries%2520%E2%80%93%2520Vol.%252074%252C%2520April%252015%252C%25202025&deliveryName=USCDC_921-DM146435

Harstad E, Hanson E, Brewster SJ, et al. (2023). Persistence of Autism Spectrum Disorder From Early Childhood Through School Age. JAMA Pediatr;177(11):1197–1205. doi:10.1001/jamapediatrics.2023.4003

Jansen, R., Maljaars, J., Zink, I., Steyaert, J., & Noens, I. (2021). The complexity of early diagnostic decision making: A follow-up study of young children with language difficulties. Autism & Developmental Language Impairments, 6. https://doi.org/10.1177/2396941520984894 (Original work published 2021)

Sandbank, M., Pustejovsky, J. E., Bottema-Beutel, K., Caldwell, N., Feldman, J. I., Crowley LaPoint, S., & Woynaroski, T. (2024). Determining Associations Between Intervention Amount and Outcomes for Young Autistic Children: A Meta-Analysis. JAMA Pediatrics. https://doi.org/10.1001/jamapediatrics.2024.1832

Author Response

Thank you for carefully reviewing the manuscript! The constructive comments are very helpful!

General concept comments
Article: While the article contains some interesting ideas the second half of the article does not read like a systematic review. The description of the literature search is understandable and appeared to follow standard search practice however, I expected more discussion of what was found in the literature search.
Review: Overall, I found this article in need of considerable work to bring it up to the quality standards of a systematic review. It also felt like it was two articles one more about a new paradigm and one a review article. As a review article there needed to be more synthesis of the results of the review. Some of the constructs in this article are interesting so you might want to do an extensive re-write. If this is the case, I encourage you to clarify the data around diagnostic assessments. For example, the age of an ASD diagnosis is now under 4 years in the US and in some states (e.g. California) it is even earlier. See reference to latest CDC report. This report just came out so you wouldn’t have had this information when writing the article. Additionally, there is data showing that children diagnosed early may no longer meet the criteria for ASD when they reach school age. (Blumberg et al., 2016; Harstad et al., 2023). Lately, I found many sentences overly long making it harder for the reader to follow I suggest looking at sentience length. 

Response: We further explained in Results about the structure of our systematic review

Specific comments: Please see my specific comments below.

39: Define neurodiversity – might not even need to use the neurodiversity construct. The article does not expand on the term later so I wouldn’t use it. If neurodiversity is important to the contract of a new diagnostic paradigm I would expand.

      Response: Added brief description of neurodiversity 

47: New prevalence rates just came out – update rates are 1 in 31 in the US.

Response: Updated in the manuscript

56: more recent studies question the need for intensive behavioral treatment (high hours) for young children (See Sandbank et al., 2024). I understand the point you are trying to make is that early diagnosis is important I think you can make that point without the McCarty and Frye quote.

Response: Cited Sandbank et al in-text 

59: update average of diagnosis with new CDC report. It has gone down.

Response: Updated in the manuscript

66: I don’t believe the DSM does provide formal assessment tools for any disorder. This statement implies that ASD is different in that regard but that is not the case.

Response: : Clarified that there are no formal tests in the DSM-5 

79: It’s not clear what transdiagnostic diagnosis really is. This is where more information about what your mean my neurodiversity could include if you decide to go in that direction. I the connection between the review of the 15 articles and transdiagnostic diagnosis was a little unclear.

Response: Brief description added 

335: Check updated screening data.

Response: Current screening data is used. 

339: Matson et al. (2012) projection that there would be fewer children diagnosed did not occur. More children have been diagnosed. Latest CDC report increasing trend of diagnosis for young children. Thus, this point doesn’t strengthen your argument.

Response: Matson’s projection is important at the time of publication. 

361: Prevalence rates and median age of diagnosis in California is 36 months, Texas is higher (see CDC report). The age range of diagnosis by state weakens this statement, because it suggests that it is not the DSM-5 criteria but differences in access to assessment by state.

Response: Matson’s projection is important at the time of publication. 

371: DSM-5 should not be categorized as “behavioral assessment.”

Response: Revision is made. 

381: Is there really wide agreement on the point of differentiation? There needs to be a citation here. I am not aware of this agreement or data that shows it is easy to differentiate.

Response: Citations are included in the manuscript. 

423: The proposed secondary assessment mentions the M-CHAT or ADOS. The M-CHAT is a screening tool (often used by pediatricians) and not in the same diagnostic category as the ADOS which is not a screening tool. Thus, this suggestion may not be something diagnosticians would do. It is also not clear what is meant by a DSM-5 based assessment?

Response: Made revisions in text and in the figure. 

435: The suggestions for biological assessment (biomarker) does not have enough support. Only one study was cited that is related to a biomarker. I don’t think we have enough data to say there is biomarker. The etiology of autism is complex.

Response: Made revisions in the manuscript. 

439: While I appreciate the intention to improve diagnostic paradigm for autism, I don’t think a strong case was made for an intergraded transdiagnostic approach.

Response: Made revisions before Table 3. 

498: Many diagnostics assessments for ASD are conducted by psychologists

Response: Revised in the paragraph. 

495: Machine learning and AI kind come out of the blue in the later part of the paper. I would suggest that if you are going to bring these up that you reference them earlier.

Response: Removed from the paragraph. 

517: A medical diagnosis can be made by non-physicians and this is done everyday

Response: Citation included in the conclusion.